# Enhancing Sustainability in Power Electronics through Regulations and Standards: A Literature Review

Li Fang [1,2,*,†], Tugce Turkbay Romano [1,3,*,†], Maud Rio [2], Julien Mélot [4] and Jean-Christophe Crébier [1]

1.  Univ. Grenoble Alpes, CNRS, Grenoble INP, G2ELAB, 38000 Grenoble, France;
    jean-christophe.crebier@g2elab.grenoble-inp.fr
2.  Univ. Grenoble Alpes, CNRS, Grenoble INP, G-SCOP, 38000 Grenoble, France; maud.rio@g-scop.eu
3.  Arts et Métiers Institut de Technologie, Université de Bordeaux, CNRS, Bordeaux INP, INRAE, I2M Bordeaux,
    F-33400 Talence, France
4.  Eaton Industries France, 38330 Grenoble, France; julienmelot@eaton.com
*   Correspondence: li.fang@grenoble-inp.fr (L.F.); tugce.turkbay@grenoble-inp.fr (T.T.R.)
†   These authors contributed equally to this work.

**Abstract:** Considering sustainability in Power Electronics (PE) is a relatively recent topic of interest. However, the existing regulatory and normative frameworks supposed to guide designers and industries in this direction have not been combined in an exhaustive way. This article aims to bridge the gap by conducting a literature review of the regulative and normative constraints for sustainability in PE. This study primarily addresses the framework at the European level, with a focus on French regulations and standards. In this study, a total of 63 relevant documents are collected and analyzed. A framework representing the overview of existing legislation and facultative guidelines for PE ecodesign is established. A collaborative online tool is developed to enable access to the inventory by PE stakeholders. The analysis of the framework outlines the limitations and challenges needing to be addressed, including the absence of constraints on environmental performance, the inadequacy of material efficiency standards for PE products, and the unclear methodology for ecodesign implementation. This work, undertaken at the European level with a detailed examination of the French context, is intended to serve as an inspiring analysis for other countries and for PE designers who are considering the regulatory framework of a European representative country.

**Keywords:** power electronics; energy-related products; sustainability; ecodesign; circular economy; state of the art; review; regulations; standards

## 1. Introduction

Power Electronics (PE) plays a pivotal role in the energy transition by facilitating the conversion of electrical attributes from renewable energy sources and meeting the demands of modern electrical loads such as E-mobilities and building air conditioning [1].

However, the lifecycle of Power Electronics impacts the environment significantly, including their high energy consumption, critical raw material consumption, freshwater use, chemical exposure and waste generation [2,3]. The sustainable development of PE products is urgently required to reduce the environmental impact throughout their lifecycle and extend their lifetime. Over the past twenty years, at the European level, the constraints and incentives imposed by regulations and standards have has a significant impact on sustainable PE product development. Providing requirements and guiding designers to develop PE products that are more durable, energy- and resource-efficient, repairable, recyclable, and with a preference for recycled materials [4].

A series of regulations and directives have been introduced with the primary goal of mitigating pollution resulting from production and end-of-life (EoL) management. These measures aim to limit or prohibit the use of toxic substances either during the manufacturing process or directly within static converters.

Notably, the RoHS Directive, first adopted in 2003 (Restriction on Hazardous Substances), and the REACH Regulation (Registration, Evaluation, Authorization, and Restriction of Chemicals), enacted in 2006 [5,6], fall within this framework. The RoHS Directive, recognized in the field of Power Electronics, led to the elimination of lead usage in Printed Circuit Board (PCB) soldering [2].

Beyond the substance level, the WEEE Directive established in 2002 governs the management of Waste from Electrical and Electronic Equipment (WEEE) in each European Union (EU) member state. A decade later, revisions were introduced to address the increasing volume of waste requiring handling. In France, consequently, both the professional and individual WEEE management systems underwent significant changes following the "Law No. 2020-105 of 10 February 2020", related to the French Anti-Waste Law for a Circular Economy ('AGEC law') [7], which profoundly transformed the framework of Extended Producer Responsibility (EPR) in France (Articles L. 541-10 to L. 541-10-17 and R. 541-86 to R. 541-178 of the Environmental Code).

At the European level, the "New Circular Economy Action Plan (CEAP) (2020)", adopted by the European Commission identified the circular economy as a crucial strategy for promoting sustainable growth in various industrial sectors [8]. In an effort to broaden the scope of the Eco-design Directive 2009/125/EC, a framework and a set of mandatory minimum requirements for energy-related products (ErPs), including energy efficiency, have been established [9]. The European Commission introduced a proposal for the "Eco-design Regulation for Sustainable Products (ESPR)" in March 2022. This proposal aims to establish ecodesign requirements to promote the sustainability, repairability, maintainability, reusability and recycling of ErP, which also encompasses PECs [10]. Currently, in France, the disassembly of electronic components is gradually becoming a part of the specifications outlined by eco-organizations responsible for facilitating the repair of a growing portion of EEE. This trend is particularly relevant given the increasing emphasis on repairability [11].

Consequently, the field of PE is profoundly influenced by these legal measures, related to the broader field of ErP. These regulations are driving PE industry towards enhanced sustainability. Simultaneously, the numerous standards arising in the circular economy perspective have been identified as being useful for framing best practices in PE.

To ensure compliance with specific criteria for requirements and the ability to adopt relevant standards for effectively implementing sustainable product development, PE designers need to comprehend the legislative and normative framework well. This understanding is crucial for defining sustainable product specifications and guiding the implementation of sustainable design.

Several studies have examined how regulations and standards impact energy-related product (ErP) sustainable design. For example, Nissen et al. (2007) presented an overview of the status of the WEEE directive and ROHS restrictions for electronics [12]. Schischke et al. (2008) discussed how the ecodesign directive regulated the energy efficiency of ErPs during their usage [13]. Hurtado Albir and Carrasco Hernández (2011) explored the impact of the ROHS and REACH regulations on limiting hazardous materials [2]. Additionally, Bundgaard et al. (2020) and Schischke et al. (2022) looked into how EN4555x contributes to designing products compliant with material efficiency requirements [14,15]. These studies are primarily concentrated on specific aspects within the legislative and normative framework for energy-related products (ErPs). However, there is currently no existing literature that offers a comprehensive overview of the legislative and normative framework for sustainable design in ErPs. Furthermore, there is a lack of illumination on how these legislative and normative frameworks are specifically tailored or adapted to the field of PE. Moreover, the precise nature of these standards, their practical utility for PE practitioners, and their limitations remain unclear. There is also a pressing need to explore how the legislative and normative frameworks can evolve to promote absolute sustainability in PE. Thus, our research question is as follows:

**How do regulations and related standards support sustainable product development of Power Electronics Converters (PECs), and what are the potential limitations of the current framework?**

This article aims to contribute to the existing literature by conducting a comprehensive literature review. Our primary objective is to investigate the regulations and standards that contribute to sustainable development in ErP, with a specific focus on their adaptation and applicability to PE. This article identifies the legislative and normative framework supporting sustainable PE product development, critically assesses the limitations inherent in this framework, and proposes potential avenues for its evolution. To achieve this, the article categorizes environmental indicators and life cycle stages addressed by the framework. Moreover, this article classifies the representative standards that facilitate the PE Product Development Process in terms of circularity evaluation, environmental impact assessment and ecodesign integration. It is important to note that this article exclusively concentrates on the framework developed in France within the context of European-level regulations.

## 2. Materials and Methods

The review was conducted using the EU Science Hub, EUR-Lex, website of The French Agency for Ecological Transition (ADEME), Ministry of Ecological Transition (France), French Standardization Association (AFNOR), International Organization for Standardization (ISO) and the International Electrotechnical Commission (IEC). Two sets of keywords were employed to search the legislative and documents at the intersection of (A) sustainable development: "eco-design" or "environment" or "circular economy" or "sustainability; and (B) "energy-related product" or "power electronics".

This analysis found 63 references, which include 6 directives, 1 law, 2 regulations, 42 standards and 12 other related references. All the references were collected, organized and analyzed using the following categories of information:

-   Agency: AFNOR, European Commission, ISO, IEC, European Country, European Committee for Electrotechnical Standardization (CENELEC), United Nations, Industry consortium.
-   Coverage: French national, European, International.
-   Type: Standards, Directive, Regulation, Law, Other.
-   Domain: General/ErP/Power Electronics.
-   Mandatory: Yes/No.
-   Life cycle stages addressed: raw material extraction, manufacturing, usage, circular scenarios (repair, reuse, remanufacturing), EoL (recycle, incineration, landfill).
-   Environmental aspects addressed: energy efficiency, waste quantity, quantity of hazardous substance, raw material consumption, material efficiency, reusability, repairability, remanufacturability, upgradability, recyclability, full environmental impact indicators (ISO 14040 [16]), PEP Ecopassport [17] (7 environmental indicators, 17 inventory flow indicators).
-   Reading priority: Priority read, Average.

Then, the gathered information was organized to establish a legislative and normative framework for ErPs, with a particular focus on the field of PE.

In order to enhance accessibility to this compilation of work, an online search tool for this inventory is provided in this paper. This tool aims to assist professionals engaged in the design and development of PE-based products by facilitating efficient search and exploration throughout the 63 references collected and analyzed.

## 3. Review of Regulative and Normative Constraints for Power Electronics Sustainable Development

According to the review method described above, the identified documents were firstly clustered into two groups: mandatory requirements and facultative guidelines. The

life cycle stages addressed and the environmental impact indicators considered by these documents are analyzed.

### 3.1. Existing Legislative Framework for PEC

The specific legislation governing the ErPs are discussed in this section. Furthermore, the manner in which regulatory provisions applying to PE products are illuminated.

The selected documents were clustered into four distinct sections based on the life cycle stage that they addressed: the raw material extraction and manufacturing of PECs, their usage, circularity options except recycling, and finally, EoL management (Figure 1).

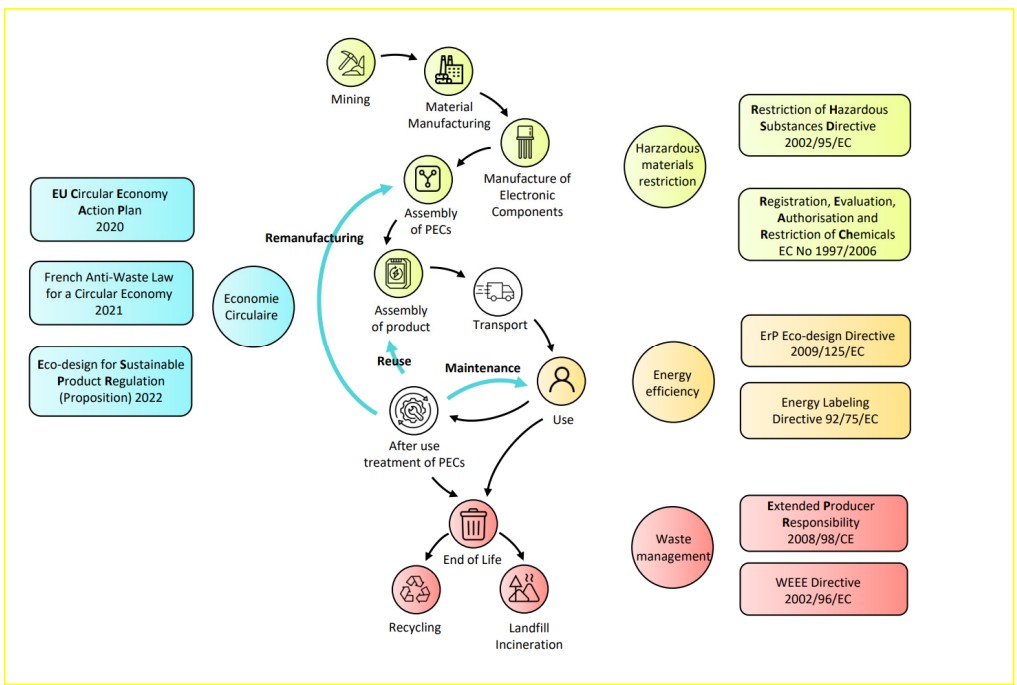

**Figure 1.** Existing sustainability policies address to Power Electronics.

### 3.1.1. Regulations on Raw Material Extraction and Manufacturing

Raw material extraction and manufacturing are the first stage of the PEC life cycle. In this stage, significant amounts of resources and energy are consumed, leading to the emissions of greenhouse gases and other pollutants [10]. The use of toxic and harmful substances in PECs can also have negative impacts on the product's EoL stage [2]. When PE-based products are discarded or abandoned, the toxic and harmful substances they contain can pollute the soil, water sources and air [2]. Many of these substances are known to persist in nature [18]. They can accumulate in the food chain, ultimately being exposed to humans [18]. To address these problems, regulations and standards have focused on the limitations of the usage of hazardous substances.

When designing PECs, it is crucial for PE designers to consider the material composition of the components and the chemical substances involved in the manufacturing process. In particular, compliance with the two EU regulations: REACH (EC 1907/2006) and ROHS (Directive 2002/95/EC) [5,6] is needed (Table 1).

REACH (Registration, Evaluation, Authorization and Restriction of Chemicals) is a European Union regulation that came into force in 2007 [5]. It is the main EU law to protect human health and the environment from the risks that can be posed by chemicals. REACH stipulates that chemical substances that exceed 1 ton per year per company must be registered with that information in a central database in the European Chemicals Agency (ECHA) [5]. The registered substances will be evaluated by the ECHA to clarify the potential risks. Authorization is required for Substances of Very High Concern (SVHCs) that are included in Appendix XIV of the REACH regulation [5].

To address the specific challenges posed by EEE, which include PECs, there is the RoHS (Restriction of Hazardous Substances) Directive that came into effect in 2006. The RoHS restricts the amount of certain hazardous substances in the manufacture of electrical and electronic products that are sold in the EU market.

Javier Hurtado Albir et al. (2011) provided a list summarizing where these toxic substances can be found in PE components [2].

- Lead (Pb) (0.1%): solder, coatings on component terminations, paints, pigments and driers, PVC as a stabilizer, batteries (not covered by the RoHS Directive).
- Mercury (Hg) (0.1%): relays.
- Cadmium (Cd) (100 ppm): electroplated coatings, special welding (e.g., in some types of fluxes), contacts, relays and electrical switches, PVC stabilizer, plastic, ceramic pigments, in some ceramic materials.
- Hexavalent chromium (CrVI) (0.1%): passivation coatings on metals, corrosion-resistant paintings.
- Polybrominated biphenyls (PBBs) (0.1%) and Polybrominated diphenyl ethers (PBDEs) (1000 ppm): plastics.

To comply with the REACH and RoHS regulations, PE designers must carefully select components and manufacturing processes that comply with the restrictions. This requires implementing appropriate measures to minimize the use of hazardous substances, as well as the use of safer alternatives wherever possible.

For instance, alternative RoHS directive-conforming components that can be used in a PEC are as follows:

- Silver oxide/cadmium contacts should be replaced by silver oxide/tin contacts, with a good performance at low voltage, although they wear faster at high voltages;
- Chrome passivation has several options, most of them are less effective as inhibitors of corrosion in uncoated metals;
- Mercury switches can be replaced by gold contact switches, but mercury offers a contact free of fluctuations, and the operative life is significantly greater;
- PBDE flame retardants can be exchanged with others, but this might clash with fire regulations. Once PBDE is used, the plastic part cannot be recycled anymore.

**Table 1.** Regulations and standards addressing the usage of hazardous substances.

| Reference | Title |
|---|---|
| RoHS directive (Directive 2002/95/EC) [6] | RoHS (Restriction of Hazardous Substances) Directive |
| REACH (EC 1907/2006) [5] | REACH (Registration, Evaluation, Authorization and Restriction of Chemicals) |

### 3.1.2. Regulations on Usage Stage

Enhancing the energy efficiency of PECs during their operation holds the key to significantly mitigating energy losses over varying usage periods, given their core function of energy conversion. Table 2 presents regulations focusing on the environmental performance of energy-related products (ErPs) during the usage stage. As a response to this imperative, the Eco-design Directive 2009/125/EC [9] has been established, setting forth mandatory minimum threshold requirements for designing Energy-using Products (EuPs), with a focal point on the energy efficiency indicator. Consequently, products failing to meet these specified thresholds face exclusion from the EU market, resulting in progressively diminished competitiveness for those demonstrating lower energy performance, as graphically depicted in Figure 2.

To encourage manufacturers to create more energy-efficient products, Directive 92/75/EC [19], requires standard information labels on seven types of household appliances (e.g., refrigerators, washing machines, dishwashers, ovens, water heaters, lighting

sources, air-conditioning appliances) showing their energy consumption and resource usage (graded from class A to G). This energy efficiency labeling enables customers to make informed choices and promotes the development of products with enhanced energy efficiency, as depicted in Figure 2 [20].

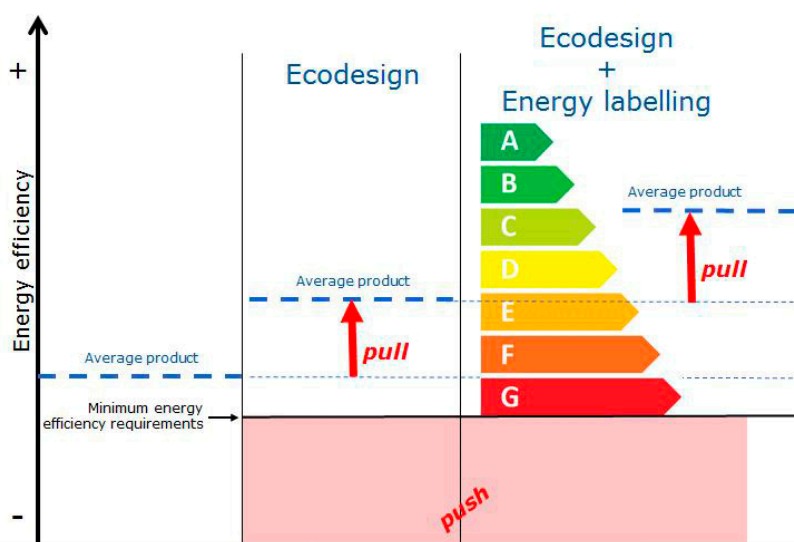

**Figure 2.** Effects of "push and pull" from minimum efficiency standards and labeling based on Commission Delegated Regulation (EU) 2017/1369 [20].

Directive 92/75/EC [19] was repealed by Directive 2010/30/EU [21] on 19 May 2010, which extend the scope to provide information on the energy consumption of energy-related products in general (e.g., lighting products, heating and cooling equipment, consumer electronics). Directive 2010/30/EU [21] was amended by Directive 2017/1369/EU [22], which introduced new labeling requirements for energy-related products. The new labeling requirements include a revised energy efficiency scale from A to G, replacing the previous A+++ to D scale [23]. The new scale is designed to be more intuitive and easier to understand for consumers. The new energy labels have been implemented in France since 2021 for five types of appliances (e.g., dishwashers, washing machines, refrigerators, blisters, televisions and screens) [24]. This dynamic "energy label" system encapsulates critical information regarding the principal functional aspects corresponding to each product category, thereby influencing energy efficiency evaluations.

**Table 2.** Regulations focusing on the environmental performance of Energy-related Products (ErPs) during the usage stage.

| Reference | Title |
|---|---|
| Directive 2009/125/CE [9] | Establishing a framework for the setting of ecodesign requirements for energy-related products |
| Directive 92/75/CEE [19] | Indication by labelling and standard product information for the consumption of energy and other resources by household appliances |
| Directive 2010/30/EU [21] | Indication by labelling and standard product information of the consumption of energy and other resources by energy-related products |
| Directive 2017/1369/EU [20] | Setting a framework for energy labelling |

### 3.1.3. Regulations around Circularity Scenario

The circular economy is an economic framework that prioritizes minimizing waste and maximizing resource usage through closed-loop systems. The concept of circularity

encompasses various scenarios, including refuse, rethink, reduce, reuse, repair, refurbish, remanufacture, repurpose, recycle and recover [25]. The regulations surrounding the circular economy aim to promote material circularity and encourage the reintroduction of the residual value of materials in the product's value chain, presented in Table 3. In 2020, the European commission published a new CEAP (2020); this action plan promotes the use of secondary materials in product manufacturing [8], and will increase their traceability.

Announced in the New Consumer Agenda and the CEAP, the "Right to Repair" is a new Directive proposal by the European Commission that aims to empower consumers with enhanced rights to repair their goods, instead of replacing them [26]. This directive seeks to tackle the barriers that hinder consumers from accessing repair services or obtaining necessary information, such as limited availability or transparency. By promoting the availability and accessibility of repair services, the directive expects to promote a circular economy and minimize waste.

To integrate the circularity requirements into the product design stage, the European Commission proposed a regulation in March 2022, establishing a framework for Eco-design Requirements for Sustainable Products (ESPR) [10]. This proposal is strategically designed to broaden the horizon of the eco-design directive 2009/125/EC. It aims to extend the ecodesign directive's scope from a singular focus on energy efficiency in the use phase to various aspects of material efficiency, such as product upgradability, repairability, maintenance and refurbishment [10].

At the French national level, the French AGEC law (Anti-Gaspillage pour une Economie Circulaire) [7], enacted in 2020, aims to reduce waste and foster a circular economy by transitioning the French economy to a more circular model, rather than relying on a linear take–make–use–dispose approach. According to the AGEC law, the French Repairability Index (FRI) [27] has been introduced to regulate EEE. This index sets a score out of ten that indicates how easily a product can be repaired. It is mandatory for several electronic and electric products, including mobile phones, computers and washing machines. However, it is not currently mandatory for PEC. The introduction of the FRI aims to encourage manufacturers to consider repairability at the design stage and make consumers aware of repair options when purchasing a device. By 2026, France aims to increase the proportion of repaired electronic and electric products from 40% to 60%. These measures ultimately promote a more sustainable approach to consumption and production [28].

**Table 3.** Regulations promoting material circularity.

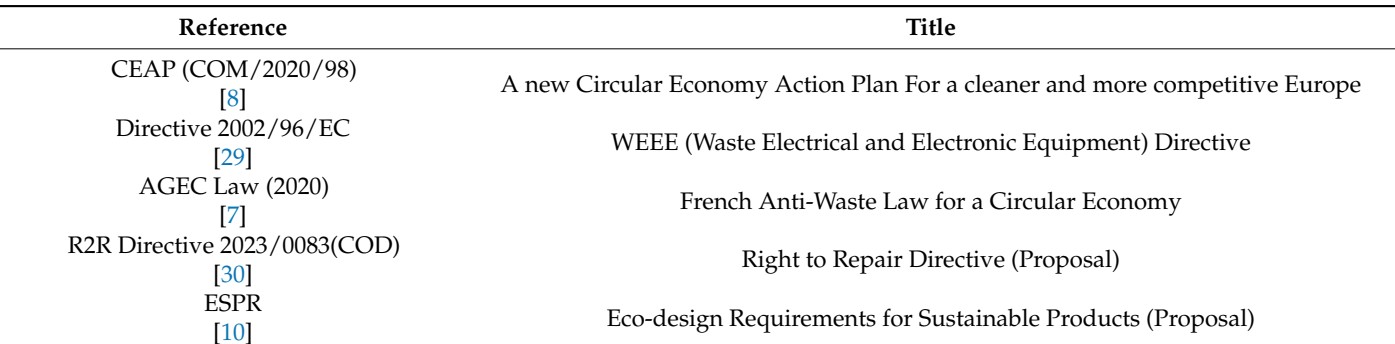

| Reference | Title |
|---|---|
| CEAP (COM/2020/98) [8] | A new Circular Economy Action Plan For a cleaner and more competitive Europe |
| Directive 2002/96/EC [29] | WEEE (Waste Electrical and Electronic Equipment) Directive |
| AGEC Law (2020) [7] | French Anti-Waste Law for a Circular Economy |
| R2R Directive 2023/0083(COD) [30] | Right to Repair Directive (Proposal) |
| ESPR [10] | Eco-design Requirements for Sustainable Products (Proposal) |

### 3.1.4. Regulations for End-of-Life Management

Initially introduced in 2003 and revised in 2012, the WEEE Directive 2012/19/EU aimed to improve the prevention, reuse, recycling and recovery of electronic waste in Europe [31] (Table 4). In 2018, the directive was revised to promote the recovery of valuable secondary raw materials, ensure efficient resource use and further reduce waste [14]. Regarding EoL treatments, PECs can contain components and substances that require special disposal or recovery procedures to prevent environmental damage. In line with the

WEEE Directive 2012/19/EU [31], the following substances, preparations or components must be disposed of or recovered separately from other WEEE:

- Capacitors (containing polychlorinated biphenyls/polychlorinated terphenyls);
- Components containing mercury;
- Batteries;
- PCBs (with a surface greater than 10 cm$^2$);
- Components containing toner, ink and liquids;
- Plastic containing brominated flame retardants;
- Components containing asbestos or asbestos waste;
- Cathode ray tubes;
- Components containing CFC, HCFC, HFC and HC;
- Gas discharge lamps;
- External electric cables;
- Components containing refractory ceramic fibers;
- Components containing radioactive substances;
- Electrolyte capacitors (height > 25 mm, diameter > 25 mm).

PECs can contain some of these components, particularly capacitors, PCBs with a surface greater than 10 cm$^2$, electrolyte capacitors and external electric cables. Depending on the design and application, they may also contain batteries at the system level. It is essential to have or to recover these components appropriately to align with the Waste Framework Directive 2008/98/EC.

One of the key principles of the WEEE Directive is the EPR (Extended Producer Responsibility), which financially obligates producers to collect, treat, recover and dispose of WEEE [32]. This incentivizes producers to design more easily reused, recycled and recovered products and encourages a circular economy approach. However, the implementation of the WEEE Directive and EPR principles varies across Member States. For instance, France has introduced its own AGEC law to encourage manufacturers to adopt more sustainable practices [7].

**Table 4.** Regulations addressing EoL management.

| Reference | Title |
| --- | --- |
| Directive 2012/19/EU [31] | WEEE (Waste Electrical and Electronic Equipment) Directive |
| EPR (2008/98/EC) [33] | Extended Producer Responsibility |

To address the need for the proper management of end-of-life disposal and the recycling of the products, many countries have established EPR schemes. Under these schemes, producers are required to financially contribute to the management of their products at their EoL, rather than relying on citizens to pay for waste collection and management through taxes. EPR schemes take various forms depending on the product category and country, but typically involve the establishment of a governing body (e.g., eco-organism in the case of France) responsible for managing the scheme and collecting fees from producers [11]. These fees are used to cover the expenses involved in the collection, transportation and treatment of WEEE, including PECs [34]. Furthermore, the AGEC Law in France aims to promote the reuse of products by creating a "solidarity reuse fund" that supports organizations and structures such as waste sorting, recovery and recycling centers that enable the reuse of products [28]. In addition, waste collection points should be set up in a way that preserves the reusability of collected e-waste, which is partially addressed in TS 50625-4 [35].

## 3.2. Facultative Guidelines for PEC Ecodesign

The mentioned legislation applies across different stages of the PEC's entire lifecycle, with a focus on environmental impact indicators, material circularity, waste management

and other relevant aspects. Notably, a significant 80% of the environmental impacts originate during the design phase [36]. This emphasizes the crucial necessity for designers to integrate these essential factors that influence product sustainability directly into the functional design and development processes. This section presents standards supporting sustainable design for PECs, categorized into three groups: material circularity assessment, environmental performance evaluation and ecodesign integration in the design process.

### 3.2.1. Material Circularity Assessment

Under CEAP, the EN 4555X series of standards were developed to provide generic material efficiency standards for ecodesign under standardization mandate M/543 [14] (Table 5). EN 45552-EN 45557 [37–40] focus on the evaluation methods for a product's ability to be used for a long time, repaired, reused, remanufactured, upgraded and recycled. EN 45558:2019 provides guidelines for critical raw material declaration, while EN 45559:2019 indicates the method for providing information related to material efficiency aspects [41,42]. The product-specific standards for PECs are currently being developed, following the guidance provided by the EN 4555X [14] series. This approach aims to align with the understanding of various material efficiency aspects within distinct product groups [14].

In addition to the EN standards, the IEC/TR 62635 [43] directives provide a comprehensive approach for calculating the recyclability and recoverability rates of electrical and electronic equipment, including PECs.

EN 50614 [44] was developed to provide requirements for preparing WEEE for reuse. This standard specifies procedures for the collection, sorting and treatment of WEEE to promote their reuse.

**Table 5.** Regulations and standards promoting material circularity.

| Reference | Title |
|---|---|
| EN 45552:2020 [37] | General method for the assessment of the durability of energy-related products |
| EN 45553:2020 [38] | General method for the assessment of the ability to remanufacture energy-related products |
| EN 45554:2020 [39] | General methods for the assessment of the ability to repair, reuse and upgrade energy-related products |
| EN 45555:2019 [40] | General methods for assessing the recyclability and recoverability of energy-related products |
| EN 45556:2019 [45] | General method for assessing the proportion of reused components in energy-related products |
| EN 45557:2020 [46] | General method for assessing the proportion of recycled material content in energy-related products |
| EN 45558:2019 [41] | General method to declare the use of critical raw materials in energy-related products |
| EN 45559:2019 [42] | Methods for providing information relating to material efficiency aspects of energy-related products |
| IEC/TR 62635 [43] | Guidelines for end-of-life information provided by manufacturers and recyclers and for recyclability rate calculation of electrical and electronic equipment |

### 3.2.2. Environmental Performance Evaluation

Standards for environmental performance evaluation allows PE designers to make informed choices and take appropriate measures to reduce or prevent negative environmental impacts. Life Cycle Analysis (LCA) is a standardized and mature systems-oriented analytical tool that provides a life cycle perspective for assessing the potential impacts of products or services [16]. The standards provide guidelines for conducting an LCA and are presented in the Table 6. The international reference program PEP Ecopassport® was developed for environmental declarations specifically for EEE products [17]. PEP Ecopassport® establishes rules for certifying type III environmental declaration (ISO14,025). PEP Ecopassport®

Program provides Product Category Rules (PCRs) and Product Specific Rules (PSRs) to conduct LCAs for electrical product families, for instance PSR0010—Uninterruptible Power Systems (UPSs) that can be used as an example for PECs [47]. There is mutual recognition between PEP Ecopassport® and other European associations that maintain the same level of data quality. Equivalent rules will be adopted at the European level when products become subject to mandatory Environmental and Social Performance Reporting (ESPR).

**Table 6.** Standards concerning environmental impact evaluation.

| Reference | Title |
| --- | --- |
| ISO 14031:2021 [48] | Environmental performance evaluation—guidelines |
| ISO 14040:2006 [16] | Life cycle assessment—principles and framework |
| ISO 14044:2006 [49] | Life cycle assessment—requirements and guidelines |
| ISO 14047:2012 [50] | Life cycle assessment—illustrative examples on how to apply ISO 14044 [49] to impact assessment situations |
| ISO 14072:2014 [51] | Life cycle assessment—requirements and guidelines for organizational life cycle assessment |
| ISO 14067:2018 [52] | Greenhouse gases—carbon footprint of products—requirements and guidelines for quantification |
| PEP ecopassport® Program [17] | International reference program for environmental declarations of products from electric, electronic and heating and cooling industries |

The current framework for conducting LCAs on PE systems (including study objectives, scope, life cycle inventory, and impact assessment) lacks the necessary precision to ensure consistent estimations among various parties. As a result, comparing products becomes challenging, especially when there are variations in the functional unit. Additionally, the selection of indicators and calculation methods for impact assessments is left to the discretion of the LCA analysts, leading to potentially significant variations in results. To address these challenges, the international working group TC111 WG15 is currently developing the IEC 63,366 standard titled, "Product category rules for LCA of electrical and electronic products and systems". The aim is to establish clearer guidelines for conducting LCA within the field of electrotechnology, ensuring a more consistent and standardized approach.

### 3.2.3. Ecodesign Integration

The mentioned criteria and evaluation system for sustainable PECs become effective only when integrated into the Product Development Process (PDP) of PECs. Table 7 presents the standards that offer guidance for aiding companies in implementing an ecodesign approach within the PDP. The ecodesign methods and tools that can assist PE designers in integrating environmental considerations alongside traditional design factors such as performance and cost [53], towards reducing environmental impacts throughout the product life cycle [54].

**Table 7.** Standards concerning ecodesign implementation in Product Development Process (PDP).

| Reference | Title |
| --- | --- |
| ISO 14006:2020 [55] | Guidelines for incorporating eco-design |
| ISO 14009:2020 [56] | Guidelines for incorporating material circulation in design and development |
| ISO 14062:2002 [54] | Integrating environmental aspects into product design and development |

Specifically for the PE field, the European standard EN 50598 [57] specifies ecodesign requirements for electrical drive systems, motor starters, Power Electronics and their applications in machine drives. Built upon the Eco-design Directive [9], EN 50598:2015 [57] is divided into three parts, outlining test and measurement methods for assessing compliance with international efficiency classes (IE classes) (Table 8). The first two parts of this standard have been repealed and replaced by the international standards IEC 61800-9 for power drive systems with variable speed drives [58,59].

**Table 8.** EN50598 Standards: ecodesign for power drive systems, motor starters, Power Electronics and their driven applications.

| Reference | Title |
|---|---|
| IEC 61800-9-1:2017 [58] | Part 1: General requirements for setting energy efficiency standards for power driven equipment using the extended product approach (EPA), and semi-analytic model (SAM) |
| IEC 61800-9-2:2017 [59] | Part 2: Energy efficiency indicators for power drive systems and motor starters |
| NF EN 50598-3:2015 (in force) [57] | Part 3: Quantitative ecodesign approach through life cycle assessment including product category rules and the content of environmental declarations |

In accordance with the "classic" ecodesign process, the standard EN 50598-3, in effect since 2015, specifies relevant requirements for PE practitioners, referring to product categories and the contents of environmental declarations related to Complete Drive Modules (CDMs) used in motor-driven applications (i.e., loads driven by motors) for low-voltage applications (less than 1000 V) with power ranges from 0.12 kW up to 1000 kW throughout their entire life cycle. This standard provides practical guidelines for Power Electronics designers to implement ecodesign principles in their activities. It starts by describing the principal elements of ecodesign, including two types of environmental declaration (Type II and Type III) (Table 9). It also defines ecodesign requirements that highlight aspects that Power Electronics designers should consider, such as the constitutive materials and substances used, energy efficiency during the use phase and potential environmental impacts during all life cycle stages, from manufacturing to EoL. The standards indicated the environmental indicators should be evaluated and declared: climate change (measured in kg $CO_2$-eq), ozone depletion (measured in kg CFC-115), the creation of photochemical ozone (measured in kg NMVOC), terrestrial acidification (measured in kg $SO_2$), freshwater eutrophication (measured in kg P), human toxicity (measured in kg 1,4-DB), terrestrial ecotoxicity (measured in kg 1,4-DB), mineral depletion (measured in kg Fe) and fossil fuel depletion (measured in kg) [57]. A CENELEC (European Committee for Electrotechnical Standardization) working group TC22X is presently in the process of updating the EN 50598-3 standard. The goal is to establish specific PE rules based on EN 63,366 as generic requirements.

**Table 9.** Standards for environmental declaration of Power Electronics-related products (EN50598:2015).

| Classification | Reference |
|---|---|
| Material declaration | EN 62474:2010 [60] |
| Conformity to applicable legislation | EN 50581:2012 [61] |
| Statements concerning the potential impacts of the product's life cycle | EN ISO 14020:2003 [62] |
| Qualitative statements on potential environmental aspects (for Type II) | EN ISO 14021:2021 [63] |
| Quantitative statements on environmental aspects (for Type III) | EN ISO 14021:2021 EN ISO 14025:2018 [63,64] |

## 4. A Collaborative Online Inventory Framework for Up-to-Date Review of Legislative and Normative Constraints

The 63 related documents contributing to legislative and normative frameworks have been gathered in an Excel file and shared online (Figure 3) (VIVAE website accessed on 5 December 2023, https://g2elab.grenoble-inp.fr/fr/recherche/projet-vivae-contenus-techniques). In the inventory, documents are characterized by parameters such as agency, coverage, mandatory status, type, domain, lifecycle addressed, environmental aspects addressed and priority for review. These parameters enable users to effectively search and evaluate relevant documents based on specific criteria.

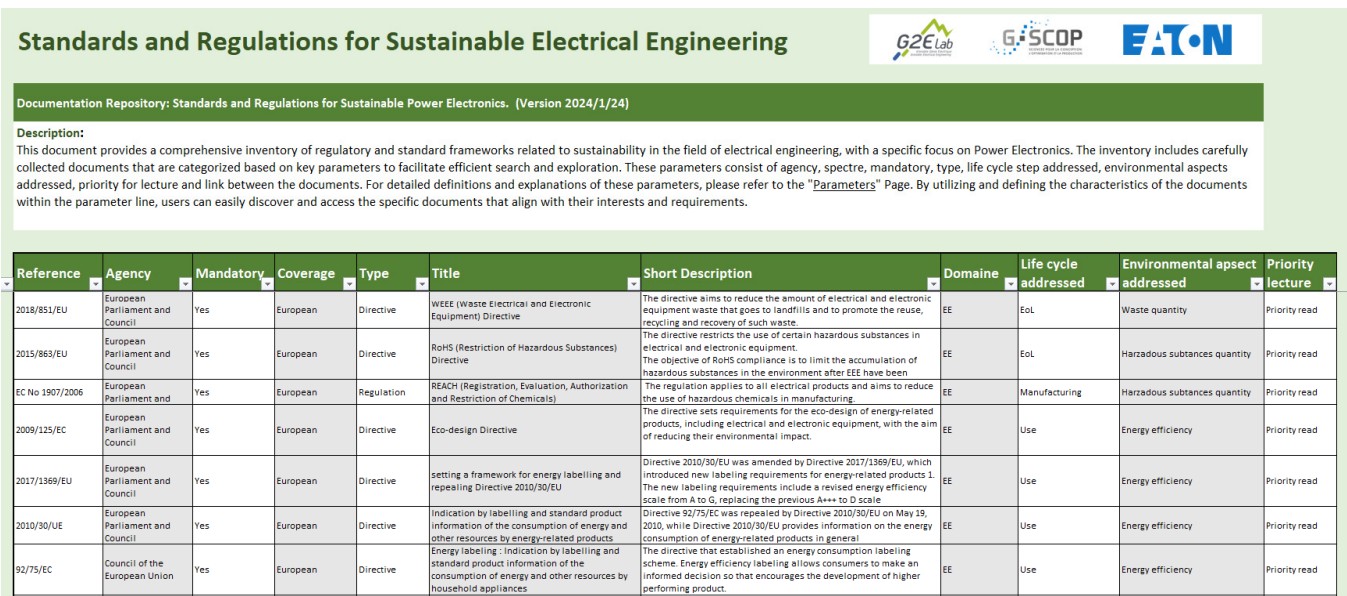

**Figure 3.** Seeker tool for analyzing current standards and regulations related to sustainable electronic engineering.

This inventory can preliminarily support professionals involved in Power Electronics-based product design and development, such as designers, R&D professionals and researchers. However, it can also support those working on sustainable development and the circular economy within the company. Although the specific PE field is the main target, any private or public organization operating in the broader domain of electrical engineering can also benefit from the documents provided in this inventory.

This inventory is designed to be flexible, meaning that the database is not fixed and can be easily updated. This flexibility enables the opportunity for contributors to aid in the development, enrichment or consolidation of the data made available to everyone. For instance, if researchers, industrialists or policy-makers come across or are developing new legislation and standards that have not yet been included in the current database, they can contribute by contacting the owners of the inventory or updating their own version, since it can be downloaded.

This seeker tool facilitates the establishment of the PE product specification, ensuring compliance with regulations and standards in the aspect of sustainability. It helps PE designers discern specific constraints applicable to the PE product specification, and determine the standardized evaluation method for analyzing these defined constraints.

The overview of environmental legislation for ErPs and PE provided by the inventory sheds light on the existing limitations within the current regulatory framework. Subsequent sections delve into a detailed discussion of these identified limitations and associated challenges.

**5. Discussion: Limitations and Challenges Related to the Current Legislative and Normative Constraints Addressed for Power Electronics Sustainable Product Development**

*5.1. Absence of Constraints on Environmental Performance*

The state of the art of the current regulatory landscape in the field of PE reveals several notable aspects. Firstly, existing standards and regulations do not impose any requirements on the characteristics and "environmental performance" of energy conversion devices beyond minimum efficiency levels and standby power consumption. These standards are primarily focused on minimizing the electrical energy dissipated during device usage [9]. The ecodesign directive has drawn criticism for its exclusive emphasis on energy efficiency during the usage phase [14]. However, it falls short in addressing material efficiency and various environmental impacts throughout a product's extended lifecycle. In the case of EN 50598-3, it only defines performance classification in terms of energy efficiency and associated power losses during the usage phase. When it comes to environmental life cycle impact indicators and EoL performance, the standard only requires manufacturers to provide relevant information.

To make the standards for assessing the environmental impact of PE systems more stringent, a shift from relative limits (which are not mandatory) to absolute limits (which would be mandatory) is suggested. Such a shift towards absolute limits would have practical implications. It could lead to removing products with the poorest environmental performance (in addition to considering energy efficiency) from the market. Moreover, based on the "absolute limits", a standardized ranking system could be set up. The ranking system would provide clear guidance to PE designers in designing products that minimize negative environmental impacts throughout their life cycle.

The absolute limits would be based on a specific set of indicators representing significant environmental effects of the PE system's life cycle. This set could align with the indicators already identified by the EN 50598-3 standard [57].

However, implementing the concept of absolute limits requires careful consideration. It necessitates a democratic discussion beforehand to determine what should be considered as a "threshold value" not to be surpassed for specific categories of PE products.

These considerations are connected to the concept of "Absolute Sustainability" in LCA [65]. While challenging, this approach involves thinking about physical boundaries beyond which Earth's ecosystems could shift into unpredictable and irreversible states. The concept of absolute limits draws from the framework of planetary boundaries [66]. Building on this framework, Bjørn et al. (2015) established limits for LCA indicators based on ecosystem capacity [65]. For instance, to keep the global average surface temperature below 2 °C, greenhouse gas emissions should remain below 985 kg of $CO_2$ equivalent per person per year [65]. Transitioning from this global limit to specific product categories while considering sector-specific boundaries requires exploring different scaling principles, all of which should be rooted in the concept of a functional unit.

*5.2. Horizontal Material Efficiency Standards Have Not Been Adapted for PE*

5.2.1. The EN4555X Standard Family Remains Generic

EN4555X standard family provides generic methods and criteria for the assessment of the durability of energy-related products, and ErPs' ability to be repaired, reused, upgraded, remanufactured, recycled and recovered.

Patra (2021), however, pointed out the lack of application of these generic horizontal standards EN4555X to unique energy-related products such as Power Electronics drives [67]. The current literature-based design guidelines are excessively general. The specific application of the EN4555X standard's circularity criteria for PECs requires a further specification and adaptation to be incorporated in the conceptual stage design requirements. The product-related and support-related criteria and their qualification criteria need to be adapted regarding different system levels of PECs. This aspect is currently being ad-

dressed by the CENELEC Working Group TC22X, aiming to adapt EN4555X to the Power Electronics (PE) system.

### 5.2.2. Limitations of the Current French Repairability Index for Supporting PE Designers

To propose to stakeholders a useful framework for evaluating product repairability, the development of a standardized and more technical repairability index for PECs becomes essential. This index would consist of distinct categories and criteria that could be assessed and given varying importance to calculate a potential repairability score. These categories would cover factors such as resilience against malfunctions, product documentation and identification, diagnostic capabilities, ease of repair, the replacement of components and requalification [68].

By taking into account elements like potential safety component integration, the simplicity of diagnostics, the availability of product information, modular design, standardized components, and the accessibility of parts, this theoretical index aims to empower stakeholders to make more informed decisions regarding repair and maintenance. Moreover, the index could potentially consider factors like the estimated time required for procedures, the complexity of component removal, the required expertise level, the availability of repair services, and the compatibility with other products.

Introducing such a standardized repairability index could potentially enhance transparency, streamline repair processes, and encourage the development of PECs that are not only more repairable but also built for long-lasting durability.

### 5.3. Unclear Methodology and Lack of Industrial Interests for Integrating Ecodesign Criteria into the PE Design Process

The current standards do not provide a comprehensive methodology to effectively integrate environmental impact assessment and material efficiency evaluation into the distinct stages of the PE design process, including tasks like conversion topology design, hardware selection, and 3D layout.

Moreover, the general ecodesign integration standards fail to specify the methodology and tools required for monitoring the relevant criteria in the context of the PE design process. This gap raises questions about the specific stage at which these criteria should be measured, what criteria should be considered, and how these criteria should be measured effectively.

This research gap poses challenges for PE professionals aiming to implement ecodesign in their R&D projects and to evaluate the ecodesign maturity of their design process. Consequently, it becomes challenging to monitor effectively and to optimize the sustainable performance of the PEC within the design process.

Moreover, the current cost-driven product development model is a barrier to convince industries to adopt ecodesign methodologies and implement ecodesign criteria in the PE design process. Striking a compromise between cost and environmental considerations becomes challenging in the absence of industrial interests.

## 6. Conclusions

This article aimed to offer the scientific community in the field of PE as well as electrical engineering a comprehensive assessment of the regulations associated with ecodesign and circular economy in Europe, with a specific focus in France.

Currently, the research is primarily focused on the broad field of ErPs. However, there is relatively limited discussion regarding how these regulatory frameworks are tailored to specific product categories. This gap is particularly noticeable in the field of PE. PECs are crucial equipment in a diverse range of electronic and electrotechnic/electrical products. However, there is an absence of comprehensive and detailed literature that investigates how sustainable regulations affect the development of PE products and how related standards contribute to promoting ecodesign for PE products. Therefore, this paper's unique contribution lies in examining sustainability regulations and standards

applicable to the PE domain, simultaneously analyzing how these standards support sustainable PE product design.

The findings present a legislative and normative framework that serves as a guide for PE practitioners seeking to integrate ecodesign practices within industrial contexts. The legislative framework encompasses regulations for ensuring compliance across various stages of the product life cycle—from raw material extraction and manufacturing to usage, circularity scenarios and EoL management. Complementing this, the normative framework establishes standardized guidelines for evaluating environmental impacts, assessing material circularity, and effectively integrating ecodesign principles. The compiled online inventory (VIVAE website accessed on 5 December 2023, https://g2elab.grenoble-inp.fr/fr/recherche/projet-vivae-contenus-techniques) (Figure 3) facilitates further exploration for PE professionals by providing an analysis tool of the existing regulatory and normative framework. This inventory provides the specifications in France. From the perspective of extending this framework to different European countries through this collaborative online inventory, the authors invite other stakeholders to incorporate the implementation of European directives into their national laws and the specific standards developed.

Through this analysis, certain gaps in regulations and standards become apparent. Firstly, the absence of constraints on environmental performance necessitates the establishment of absolute limits for LCA indicators and a ranking system to address this shortcoming. Secondly, while horizontal material efficiency standards exist, they require adaptation with suitable granularity to align with the specific needs of PE industrial design. Thirdly, the PE design process lacks a coherent methodology for integrating ecodesign criteria into the activities of PE designers. Lastly, an incentive regulatory method is required to stimulate the industry to actively integrate ecodesign requirements. To tackle these challenges, it is crucial for general standards to incorporate the industrial design context of PE. By doing so, relevant tools can be developed, allowing stakeholders to assess the maturity of their ecodesign process and effectively monitor the sustainable performance of their products during the design phase.

**Author Contributions:** Conceptualization, L.F., T.T.R., M.R. and J.-C.C.; methodology, L.F., T.T.R., M.R. and J.-C.C.; validation, L.F., T.T.R., M.R., J.M. and J.-C.C.; formal analysis, L.F., T.T.R., M.R., J.M. and J.-C.C.; data curation, L.F. and T.T.R.; writing—original draft preparation, L.F. and T.T.R.; writing—review and editing, L.F., T.T.R., M.R., J.M. and J.-C.C.; visualization, L.F. and T.T.R.; supervision, M.R. and J.-C.C.; funding acquisition, M.R., J.M. and J.-C.C. All authors have read and agreed to the published version of the manuscript.

**Funding:** This research was funded by the French National Research Agency (ANR) under the project ANR-21-CE10-0010-01.

**Data Availability Statement:** The data presented in this study are available at https://g2elab.grenoble-inp.fr/fr/recherche/projet-vivae-contenus-techniques (accessed on 23 January, 2024).

**Conflicts of Interest:** Author Julien Melot was employed by the company Eaton Industries France. The remaining authors declare that the research was conducted in the absence of any commercial or financial relationships that could be construed as a potential conflict of interest.

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
