# Peer review of "Enhancing Sustainability in Power Electronics through Regulations and Standards: A Literature Review"

_sustainability, doi:10.3390/su16031042_

Round 1

Reviewer 1 Report

Comments and Suggestions for Authors

Dear authors,

I would like to congratulate you on your manuscript. The topic is very pertinent and approached in a very complete and interesting way. I recognise the enormous amount of work that has been done and congratulate you on the work presented in the manuscript. The writing is fluid, and the work is very well structured.

Below are some comments and suggestions that I think could improve the manuscript.

Good luck with the article,

Best regards,

1. According to the journal's template, surnames should be written in lower case, except for the first letter, of course.

2. Line 31: I would advise the authors to change the beginning of the section "As we recognise", or even remove this part, as it is not very formal for the beginning of a section.

3. In the Introduction section, a review of scientific work published in the same area should be included, to show the gap that the authors intend to fill with this work. Are there similar works, even if they were carried out in another country? Are there any articles compiling legislation on power electronics?

4. In line 104, the authors state that they found 63 references; however, in line 127, they mention 61. I would ask the authors to review this aspect.

5. Line 128: The authors have "3. 3. Regulative", but I think it's just 3. Regulative.

6. Figure 1: The figure is very good; however, I would ask the authors if you could remove some red underlining from some parts.

7. Lines 155 and 157: There seems to be a lot of spacing after the word "REACH".

8. There are some tables for which there is no reference in the text, namely Tables 1, 2, 3, 4, 5, and 8.

9. Table 2: "Directive 2010/30/UE". This is just to alert you to a small oversight in the swapping of the letters E and U.

10. Line 254: Perhaps capitalise the law, as it is associated with a specific one.

11. Line 280: It must have been an oversight, but I wanted to warn the authors to put the 2 in superscript in the cm.

12. Leave a small spacing before line 291 and line 304.

13. Table 7: The numbering of line "386" appears within the Table.

14. Table 8: The heading is not bolded as in the previous ones. I have a question about this table because the content doesn't seem to be linked to the text that precedes it.

15. Page 12: Where Figure 2 appears, it should be: Figure 3.

16. In Section 5.2.1 the EN4555X family of standards appears with the "x" in lower case; however, on page 9 it appears in upper case. I would ask the authors to review this aspect. Furthermore, on Line 480, "N" is in lower case. On Line 491 there is a space between EN and 4555 which does not exist on the others.

17. Line 619: I think reference 18 is incomplete.

18. Line 657: Indicates a correction.

Reviewer 2 Report

Comments and Suggestions for Authors

This paper is well written and organized. They developed a comprehensive and significant analysis for the recent regulations and standards related to the PEC. Only some suggestions are listed as follows:

(1) What is the meaning or definition of PE sustainability? The motivation is to construct the proper regulation and standard for PE sustainability. But the authors should further clarify the direct/straightforward relationship between these regulation and standard with PE sustainability.

(2) What is "PE conversion system"? Is it some other words for PEC? Or the regulation and standard?

(3) If the word "VIVAE" is the abbr., please provide the whole words.

(4) A reduction of "3." is used for the Section 3.

Author Response

Dear reviewer,

Thank you very much for your encouraging feedback and congratulations on reviewing our manuscript. We appreciate your positive remarks about the relevance of the topic and the comprehensive approach we have taken. Your recognition of the substantial effort invested in this work is truly motivating.

We are grateful for your constructive comments and suggestions to enhance the overall quality of the manuscript.

Please find attached our response to your revisions.

In response to your comments, efforts have been made to address the issues raised, as outlined in the responses to your general concept comments and specific suggestions. The revisions are marked in yellow within the revised article.

Thank you again for your valuable input, and we hope the revisions address your concerns satisfactorily.

Best regards,

Reviewer 3 Report

Comments and Suggestions for Authors

Author Response

Dear reviewer,

Thanks for your attentive review of our paper and the constructive suggestions to improve the article. We appreciate your thoughtful feedback.  

Please find attached our response to your revisions.

In response to your comments, we have made significant efforts to address the issues raised, as outlined in the following responses to your general concept comments and specific suggestions. The revisions are marked in green within the revised article.

Thank you again for your valuable input, and we hope the revisions address your concerns satisfactorily.

Best regards,

Round 2

Reviewer 3 Report

Comments and Suggestions for Authors

I believe the manuscript has been sufficiently improved.